# Solutions-Based Approach to Urban Cat Management—Case Studies of a One Welfare Approach to Urban Cat Management

**DOI:** 10.3390/ani13213423

**Published:** 2023-11-05

**Authors:** Caitlin Crawford, Jacquie Rand, Vanessa Rohlf, Rebekah Scotney, Pauleen Bennett

**Affiliations:** 1School of Biological Sciences, Queen’s University Belfast, Belfast BT9 5DL, UK; 2Australian Pet Welfare Foundation, Kenmore, QLD 4064, Australia; 3School of Veterinary Science, The University of Queensland, Gatton, QLD 4343, Australia; 4School of Psychology and Public Health, La Trobe University, Bendigo, VIC 3552, Australia

**Keywords:** free-roaming cats, One Welfare, urban cat management, semi-owner, pound, shelter, euthanasia, sterilization, Australia

## Abstract

**Simple Summary:**

When multiple urban free-roaming cats are being cared for by an owner or a cat caregiver, it often results in public complaints due to concerns about nuisance behaviors, their effect on wildlife, and human and pet health. The typical response from the authorities is to implement a cat management strategy based on trap–adopt or kill. Because the cats are often timid or shy, many are euthanized. This strategy is detrimental to the well-being of the people who care for urban cats, and is not effective at reducing the free-roaming cat population. This qualitative study aimed to explore the impacts of a free sterilization, microchipping, and vaccination program on the people who care for multiple urban free-roaming cats. Several main themes arose during the interviews. The results demonstrate the strong bond the participants had with the cats they cared for, and the positive impact the free sterilization program had on the cat carers’ well-being and quality of life. It is recommended that this care-centered approach be used instead of current lethal cat management strategies, to improve the well-being of people and cats, reduce the free-roaming cat population, and minimize public complaints and cat impoundments.

**Abstract:**

Urban free-roaming cats create concern about their impacts on wildlife and human health, leading to the use of trap–adopt–kill methods to manage these populations. This method is ineffective at decreasing the free-roaming cat population and has a negative impact on cat caregivers’ well-being. Using semi-structured interviews, this study explored the relationship that semi-owners (people who feed cats but do not perceive ownership) and owners of multiple cats have with the cats they care for, and the social and psychological impacts of an alternative assistive-centered approach to urban cat management. This approach to semi-owned and owned cats provided free sterilization and preventative healthcare. Our findings demonstrate that the caregivers had a strong emotional bond with the cats they cared for. The caregivers also experienced a positive impact on their quality of life, and indicated an improvement in the cats’ welfare after having the cats sterilized through this program. Additionally, the cat caregivers indicated that they had a negative view of agencies, such as the municipal council. It is recommended that an assistive-centered approach to urban cat management be prioritized by local councils and welfare agencies to improve caregivers’ quality of life and psychological well-being, whilst also improving cat welfare. The implementation of this assistive-centered management approach could improve the relationship between communities and the agencies involved, leading to the continuous reporting of free-roaming cats for sterilization. This assistive-centered approach has the potential to reduce the free-roaming cat population, their effects on wildlife, nuisance complaints, and council impoundments, and is aligned with the One Welfare philosophy.

## 1. Introduction

Over the last 2000 years, free-roaming cats (*Felis catus*) have been actively transported to almost every part of the world as companions or for rodent control [1], and have successfully established themselves due to their high fecundity, versatile diet, and range of habitat [2]. A large population of free-roaming cats creates concerns about their effects on wildlife populations and human health and, in some cases, the cats’ welfare [3,4,5]. In Australia, cats are categorized as feral or domestic, with feral cats having no reliance on humans, and living and reproducing in the wild, separate from humans [6,7,8]. Domestic cats live in the vicinity of humans and depend wholly or partially on humans for survival. Free-roaming cats in urban and peri-urban areas and around farm buildings are domestic cats, and are either owned, semi-owned (fed by one or more people who do not perceive ownership), or unowned (obtain food from humans unintentionally). If unidentified with a microchip or collar and tag, they are considered stray cats [6,7,8]. In Australia, fears of wildlife predation and complaints about nuisance behaviors, such as urinating, defecating, and the noise caused by fighting, often take priority over cats’ welfare [2,9,10]. This has led to a focus on lethal enforcement-centered management. The standard management strategy in Australia is trap–adopt–kill, whereby free-roaming cats are trapped, held for a mandated holding period (usually 3–8 days) and, if not reclaimed by the owner, the socialized cats are adopted or transferred to a rehoming organization and the remainder are euthanized [11]. Stressed or fearful cats that appear timid, shy, aggressive, or unfriendly are frequently deemed feral and are euthanized without being given time to acclimate to the shelter environment, which may take eight or more days [12,13]. Applying the word feral to free-roaming domestic cat populations happens regularly and is problematic, as it makes lethal management methods more palatable by devaluing the cat’s life. This implies that these animals can be treated differently from those that are deemed to be pets [14]. This trap–adopt–kill management leads to the unnecessary euthanasia of healthy and treatable cats and kittens. In Australia, from 2018 to 2019, 33% of the cats that entered shelters and municipal animal facilities were euthanized [15].

### 1.1. Effects of Enforcement-Centered Management on Human Well-Being

Shelter staff performing the euthanasia of healthy and treatable animals experience negative psychological impacts and higher rates of suicide [16,17,18]. Shelter employees are five times more likely to show symptoms of post-traumatic stress disorder than the US national average [18], providing further evidence that lethal enforcement-centered management has detrimental physical and psychological effects on human well-being [16]. Recently, the negative impact of lethal enforcement-centered management on cat semi-owners (cat caregivers) has been documented [19]. Semi-owners are people who feed or provide care for cats, but do not perceive that they own them, and are also referred to as cat caregivers [6,20]. Cat semi-owners feel sympathy and affection for stray cats [21], suggesting that they are compassionate people who care for cats with the intention of improving the cats’ welfare. These semi-owners have been shown to have similar levels of attachment to the cats they care for as owners do for their pets [22]. Additionally, semi-owners give their time and money to care for these cats, typically providing food once or twice a day, and health care when necessary [23]. At the Port of Newcastle in New South Wales, Australia, cats that had been cared for by semi-owners for up to 18 years were culled by shooting [19]. The semi-owners described the immediate impact of the cull as “traumatic” and “horrific”. The long-term impact on the semi-owners’ well-being was evident 12 months after the cull, with some having to take a leave from work and others being unable to eat normally due to the stress [19]. Many cat semi-owners have a distrust of authorities regarding the management of stray cats, due to concerns that the cats will be killed [19,24]. This distrust, and other issues like cost [25], create barriers for cat semi-owners to access support, such as sterilization. These barriers may be the reason that semi-owned cats are less likely to be sterilized and more likely to have a litter of kittens than owned cats [26].

### 1.2. Public Opinion

Due to concerns over the euthanasia of healthy animals, lethal management strategies have failed to gain public support [9]. In New Zealand, only 23% of survey respondents supported lethal management strategies [27]. Similarly, in Brisbane, only 28% of respondents supported the lethal management of urban stray cat populations; this number decreased further (18%) when respondents were given information about non-lethal alternatives [9]. These studies highlight the need to move towards non-lethal alternatives. In countries which have similar attitudes towards companion animals, assistive-centered management strategies are favored by the public, with 90% of survey respondents in Ontario, Canada, supporting responsible pet ownership education, and 86% favoring low-cost spay/neuter programs [28].

### 1.3. Effectiveness of the Current Management Strategy and a One Welfare Alternative

One Welfare is a philosophy which describes the links between animal welfare, human welfare, and environmental conservation and sustainability [29,30]. The current lethal enforcement-centered management strategy is ineffective in the long term at reducing the free-roaming cat population in cities and towns, is not cost-effective, and is unacceptable to a large portion of people [31,32]. This suggests that this management strategy is ineffective at reducing the impact that free-roaming cats have on wildlife and the environment, and is costly to the municipal pounds and shelters that implement this strategy. Additionally, this management strategy has a negative impact on the psychological well-being of shelter workers and cat semi-owners [16,17,18,19]. Therefore, lethal enforcement-centered management fails to encompass a One Welfare approach, and instead results in negative impacts on people, animals, and the environment. In contrast, an assistive-centered management strategy, which aims to help owners and semi-owners care for their animals and is based on the sterilization of free-roaming cats, acknowledges the mutual dependency of animals, humans, and the environment, and has been documented to reduce shelter intake and euthanasia [33]. Moreover, community-based trap–neuter–return (TNR) has been shown to lower cat intake and euthanasia at shelters and municipal pounds by reducing the number of kittens born. Return-to-field (RTF), where cats in shelters and pounds that cannot be readily adopted are sterilized, ear-tipped, and returned to the location where they were found by animal shelters, decreases the euthanasia of timid and shy cats [25,34,35]. Supporting cat semi-owners in caring for cats through assistive-centered management could improve the welfare of semi-owners and the cats they care for. Additionally, this strategy could help to decrease the environmental impact cats have, by reducing their numbers through preventing kittens being born [31,36]. Assistive-centered cat management is aligned with a One Welfare approach, which aims to balance and optimize the well-being of people and animals, whilst also protecting wildlife and the environment.

### 1.4. Gaps and Aims

Whilst the negative psychological impact of lethal enforcement-centered urban cat management on semi-owners has been documented [19], the positive impacts of an assistive-centered urban cat management strategy on semi-owners’ mental well-being have yet to be investigated. People caring for multiple cats were selected for this study because the authorities frequently use lethal enforcement-centered management in an attempt to mitigate cat-related issues [37]. This involves the repeated trapping and removal of cats, which are typically euthanized because they are often timid and shy, and are frequently deemed “feral” if they show normal fearful behaviors, such as growling, striking, or hissing shortly after trapping. Domestic cats can require, on average, anywhere between 5 days and 5 weeks to acclimatize to a new environment and stop showing fearful behaviors [38]. Fines and even jail sentences may be given for feeding stray cats [37]. Less frequently, when multiple cats are being cared for on public property, other lethal methods, such as shooting, are employed [19].

This study aimed to use qualitative methodology to investigate, firstly, the relationship between semi-owners caring for multiple cats and the cats they care for. Secondly, we aimed to investigate the perceived impacts on cat caregivers prior to and after the implementation of an assistive-centered program based on free sterilization, microchipping, and preventative veterinary care for their cats. Specifically, we aimed to investigate the perceived impacts on cat caregivers’ quality of life, cat welfare, nuisance behaviors, support from agencies, and social capital (social relations that produce individual and collective benefits [39]). Thirdly, we aimed to investigate what caregivers thought might have happened if they had not received this assistance.

If an assistive-centered approach to cat management is documented to be beneficial to cat semi-owners’ psychological well-being and quality of life, it would provide further evidence to local governments and welfare agencies of the beneficial impacts of alternative urban cat management strategies.

## 2. Materials and Methods

### 2.1. Research Design

This study used a phenomenological approach to understand and explore the lived experience of people who have had the cats they care for sterilized through a Community Cat Program. In this program, cats being cared for by cat caregivers (semi-owners) and owners were provided with free sterilization, microchipping, vaccination, endo- and ectoparasite control, and veterinary care for the issues affecting the welfare of their cats. The program is an initiative of the Australian Pet Welfare Foundation in collaboration with the Royal Society for Prevention of Cruelty to Animals (RSPCA) Queensland, and the Animal Welfare League Queensland, with funding and in-kind support from multiple organizations, including the Fondation Brigette Bardot for sterilizations and MSD Animal Health for vaccinations and parasite control. The cats that were not owned were desexed, microchipped, and ear-tipped as Restricted Matter (approved by the Queensland Government under a Department of Agriculture and Fisheries Scientific Research Permit No. PRID000825). Animal Ethics Approval (Permit Number 2019/AE000207) from the University of Queensland’s Research Ethics and Integrity Unit covered the cats in this study. For the semi-owned cats, when there was no owner registered on the microchip, the cat was registered as <suburb name> Community Cat, with the Australian Pet Welfare Foundation phone number listed as the secondary contact.

A phenomenological approach allows the researcher to understand a phenomenon from the perspective of the people involved [40]. “The phenomenon dictates the method (not vice-versa) including even the type of participants” [41] (p. 294). Thus, this is an appropriate methodological approach with which to investigate the thoughts and feelings of cat semi-owners regarding an assistive-centered cat management strategy. The population of this study were semi-owners and owners caring for multiple cats located in the city of Ipswich, Queensland. Semi-structured interviews were conducted by the first author (CC) and were used to enable a deeper understanding of the social and psychological outcomes of assistive-centered cat management on cat semi-owners.

### 2.2. Participants

Due to the nature of this study, the participants were recruited through a targeted process, whereby owners and semi-owners caring for multiple cats who had had their cats sterilized through the Community Cat Program were approached to participate in this study. The participants were all over the age of 18, residents of Ipswich City, and were recruited via phone and email. Their contact details were obtained via the Community Liaison Officer for the Community Cat Program and the Cat Assistance Team (C.A.T) Coordinator. A total of 12 participants were recruited, with 10 interviews taking place between 9 June 2023 and 20 July 2023. One interview was excluded for not meeting the criteria of caring for multiple cats. Four participants were interviewed during two of the interviews; the remaining interviews only had one participant. Of the interviews used for analysis, seven of the participants identified as female and the remaining four identified as male; 10 of the participants were estimated to be in middle-to-late adulthood, with one participant estimated to be in early adulthood. One of the interviews was conducted on a farm, one interview was conducted at a business, and the remaining interviews were at private residences. At the time of the interviews, the participants were caring for between 3 and 16 cats, with a median of 7 cats. Of these, three households had taken ownership of the cats (4–7 cats), five did not own the cats they cared for (3–16 cats), and one had a mix of owned (4 cats) and unowned cats (3 cats).

### 2.3. Data Collection

Ethics approval was obtained from the University of Queensland Human Ethics Committee (2023/HE000587) before beginning this study. This study used purposive sampling [42] to reach the owners and semi-owners who cared for multiple cats and had had their cats sterilized through the Community Cat Program. Potential participants were contacted via phone and email and invited to participate in this study. Once the participants indicated interest in taking part, they were provided with a participant information sheet (PIS) and a consent form via email, or as a printed document. The PIS informed the individuals that participation in this study was voluntary, they did not have to answer any questions they did not feel comfortable with, and they could withdraw from the study at any time. The participants were also informed that any information they provided was confidential, and any information that would disclose their identity would not be published without their consent. The PIS also indicated that during the interview, participants could be asked to discuss uncomfortable memories, and the contact information for three different counselling support lines was provided in case the participants should require any assistance or emotional support.

The data were collected using semi-structured in-depth interviews. The interviews were conducted in person and voice recorded using an Olympus Voice Recorder VN-541PC. Before beginning the interviews, the participants were read the PIS and given the consent form to sign. The interviews lasted between 38 min and 83 min (average 55 min). The questions focused on the participants’ relationships with the cats they cared for, their experience before receiving support from the Community Cat Program, their experience after receiving support from the Community Cat Program, and what they thought would have happened if they had not received support from the Community Cat Program. Once all the interviews were completed, eight were transcribed by the first author (CC) and two, which had multiple interviewees, were transcribed by a professional transcription service (Pacific Transcription Pty Ltd., Brisbane, Australia). The recordings and transcripts were kept private and were saved in secure password-protected files at the Australian Pet Welfare Foundation. The transcripts were then analyzed using thematic analysis to search for recurring words or units of meaning and organized into groups and themes [43]. This was performed by coding the transcripts and organizing these into corresponding themes. The interpretation of the data was then discussed amongst the research team until a consensus was reached. All the human names were omitted, and the cat names were changed during the write-up of the results to maintain the confidentiality and privacy of participants.

## 3. Results

Several main themes and sub-themes were extracted from the interview transcripts as a result of the thematic analysis. These themes and sub-themes are discussed in the following results section and have been tabulated with context examples (see Table 1).

### 3.1. Relationship with Cats

The semi-owners and owners (caregivers) indicated that they cared deeply for the cats they were feeding and had strong emotional bonds with them. When asked to discuss their relationship with the cats, the interviewees described the cats as part of the family and likened them to having children. They used words such as “love” and “my babies”. The majority of cat caregivers saw the cats as their cats, even if they did not take official ownership. The human–animal bond is illustrated in the quotes below:


*“I love them. I love them, and he loves his cats, you know.”*



*“I wouldn’t be without them.”*



*“They are the reason, I get up.”*



*”They’re my babies, sort of like they’re my kids.”*


Additionally, several responses showed that the cats had a positive effect on the caregivers’ well-being. The caregivers described the cats as calming, discussed finding comfort in the cats’ presence when feeling unwell, and finding joy in talking about the cats with other people. This positive impact on the carers mental well-being is described in the caregivers’ own words below:


*“I get joy from hanging out with them, and sometimes particularly if I feel unwell, (…) we’ll have a pat and that helps.”*



*”[If] I was having a bad day and if I was to talk about them my moods just lifted. (…) I know if I’m really down, I’ll actually come out here and cuddle them…”*


Whilst all the caregivers were providing food for the cats, many of the caregivers were also spending a significant amount of money on caring for the cats, by buying materials with which to build outdoor enclosures, providing veterinary care to treat injuries and illnesses, and some buying cameras to monitor the cats’ welfare. These purchases were all made despite the participants living in low-income suburbs. This investment is shown in the excerpts below:


*“I’ve probably spent… I think I stopped calculating at about 1600 [AUD], with the crate, the traps, the security cameras, the palace, the fairy lights for the palace, the toys, and that doesn’t include the food or the kitty litter.”*



*“He’s been on antibiotics, and we spent about AU $3000 on him for the hospital.”*


Some of the cat caregivers also indicated that they felt a sense of responsibility or duty to the cats and the overall situation surrounding issues with the free-roaming cat population, with some mentioning that they wanted to do “the right thing”. This is shown by their responses during the interview in the excerpts below:


*“They’re here, somebody’s gotta do something about it (…) So I guess we just took on the responsibility of how do we sort these cats out.”*



*“Even if I don’t have that ownership, it’s like a bit of a duty to them.”*


### 3.2. Before the Community Cat Program

The participants were asked several questions about their situation before becoming involved with the Community Cat Program. The analysis of the responses to these questions revealed four main themes.

#### 3.2.1. Human Quality of Life

When given the opportunity to talk about the issues the cat caregivers had before receiving support from the Community Cat Program, the responses suggested a lower level of quality of life. The responses indicated strains on their social relationships, feelings of powerlessness, and a lack of knowledge on how to best deal with all of the cat-related issues. As well as this, the caregivers indicated two further themes which are linked to human quality of life: worries about cat welfare and cost, and cat nuisance behaviors.

The strains put on social relationships, the feelings of powerlessness, and the lack of knowledge are shown through quotes from the transcripts:


*“Having so many cats in the yard, it did start affecting my husband and my relationship a little bit. (…) We kind of had a few little tiffs about how to manage it.”*



*“That sense, not quite hopelessness, but (…) this sense that there’s a problem and not feeling that I had the power to do something about it.”*



*“So, yeah, because I didn’t know who to turn to before. I thought, well, I’m not going back there [the pound], and if they come here again and I’m going to get a fine. It was very, very stressful. Yeah. I mean, our health isn’t the best, so, I mean, we didn’t want that extra stress.”*


Caring for the cats often led to worries about the cost of caring for them, and many of the participants indicated that they could not afford veterinary care, such as desexing, in addition to the cost of feeding the cats. This is articulated in the participants’ own words below:


*“It was costing us a fortune, it was getting too much.”*



*“…by the time I took him down for desexing they went his eye needs to come out (…) I was at that point going shit I’m going to have to surrender him because I can’t afford eye surgery on the cat (…) [the Community Cat Program] were very good, they put him through and got his eye sorted for us.”*



*“Couldn’t afford the veterinary care. Yeah, yeah. No way. Everyone [veterinarians] around here, the general quote was 350 to 400 dollars to get a female desexed, the males were about 280 to 300 and considering there were so many, there was no way I could do that. So, I just couldn’t afford that, and feed them all at the same time.”*


#### 3.2.2. Nuisance Behaviors

Problems arising from cat nuisance behaviors were a strong theme in several of the interviews, with words such as “noise”, “feces”, and “unwanted litters”, frequently mentioned. These nuisance behaviors often led to a decrease in quality of life, as they affected the caregivers’ environment, sleep habits, and created worries about the health implications. The nuisance behaviors and their impacts are relayed in the caregivers’ own words:


*“I mean they spray, and they poop, and they smell, (…) you’re doing something [at home] and then all of a sudden there’s a whole lot of cat poop on you.”*



*“…they have fights at night, and that sets off our dog.”*



*“It was very destructive to the property. Like my daughter couldn’t really play outside without stepping in something. (…) You couldn’t really walk barefoot anywhere [because] like there’s a health implication with so much feces around.”*



*“We did have a flea problem at one stage.”*


#### 3.2.3. Cat Welfare

The interview data suggested that the welfare of the cats being cared for was poor before receiving support from the Community Cat Program. The cat caregivers discussed the poor condition the cats were in, the injuries they had, and that the cats did not show play behaviors. These indicators of poor welfare are evidenced in the following quotes:


*“They were skinny and their stomachs were shrunk in, and they looked—you know, like they were tired.”*



*“It made me upset, the fact that I could hear them fighting and when they got desexed and brought back, they actually told us that they had a lot of scarring because of the fighting…”*



*“She doesn’t know how to play with toys, she’s never had toys. Like I bring out toys and she gets scared whereas the kittens [are] going nuts. She doesn’t know how to play with other cats.”*


The caregivers also expressed concern about the safety of the cats before receiving support from the Community Cat Program. These concerns generally motivated them to start caring for the cats. This is shown in the quotes below:


*“The kittens could have been killed by the dogs getting into the wrong yard, run over by cars. (…) It’s just about them getting hurt and being hungry and not being loved and out in the weather, out in the cold.”*



*“We have not so nice neighbors in the junkyard, (…) we were very concerned that if he got his hands on any of them, the cats wouldn’t live.”*


#### 3.2.4. Perception of Support from Agencies

When asked about whether they had reached out for support before engaging with the Community Cat Program, the semi-owners and owners indicated three main sub-themes: a perceived lack of support from the authorities, negative interactions with the authorities, and a fear for the outcomes of the cats. These are demonstrated in the quotes below:


*“It was more like, can we get some traps? Can we get some help with this? And they [the council] were just like they were so blasé about it. They didn’t really care, and they didn’t understand just how bad the problem was.”*



*“I didn’t know what my chances were, after that conversation with the council, what my chances were of catching them, how hard it would be. I suppose it still felt out of my grasp to do something.”*


The semi-owners who dealt closely with the council regarding complaints relayed that they had had negative interactions with the authorities, and that these had led to feelings of anger and fear. The following excerpts provide more context:


*“I was worried. I was petrified, actually, that I was going to get fined and all this carry on. She [the council] said, well, when I come back here and I see you feeding them again, she said, I will get rid of all your cats. (…) I felt really angry.”*



*“Because I used to dread when I came home on an afternoon, and I would be looking for a van out there. I thought, oh, no. It just got to me, you know, that every time I pull up over here, they’ll [the council] be waiting for me.”*



*“The council says, oh, can you go up and touch them? I said, I can. I said, they come to us, but strangers? No. They’re feral, she said, they’re feral. I said, they are not feral. I can go out there now and I can run my hand right down the backs of them. I thought, why is she calling them feral?”*


Several of the interview participants spoke about how they did not want to engage with the authorities and worried about nuisance complaints from neighbors, as they believed that the cats would be taken away and euthanized. The quotes below provide some insight into the participants’ perception of the outcomes for the cats:


*“I don’t want to engage the Council until yeah, I… I know the answers to those questions, because if they turn around and say, well, you’ve got too many cats, you gotta get rid of them, then what do I do? Yeah, it’s my fault then. (…) [They say] so you’ve gotta get rid of them, and they’ve gotta go to the RSPCA, and then my worst nightmare happens, that Molly gets put down because she’s antisocial.”*



*“I was just really worried that they were going to go to sleep [euthanized] and not be in my backyard.”*



*“I didn’t want them [the council] to take my cats away.”*



*“I don’t want them to have a bad ending, yeah, through a council process, I suppose.”*


### 3.3. After the Community Cat Program

The semi-owners and owners were asked a series of questions to explore their experience after receiving support from the Community Cat Program. The analysis of the responses to these questions extracted four main themes, which are shown in the following paragraphs.

#### 3.3.1. Human Quality of Life

The cat caregivers’ responses appeared indicative of an improvement in their quality of life after receiving support from the program. Specifically, they explained that they felt less worried, more empowered, had an improved relationship with their cats, and felt a sense of pride and fulfilment. Additionally, the theme of social capital was found during the analysis of the transcripts, which is strongly associated with quality of life.

The caregivers discussed that after receiving support from the program they felt less worried about several things: fighting, unwanted litters, the cats being removed by the council, and their own health and well-being. Words and phrases, such as “less stressful”, “less concerned”, and “relieved” were used in the responses. The carers’ thoughts and feelings are given more context in the following extracts:


*“We can sit out the back and lay in the grass and they can cuddle up with us. So, we don’t have to worry about getting sick doing that.”*



*“I feel really good. Yep. Really good. I feel less stressful, I should say.”*



*“[I’m] relieved (…) because I know they can’t have babies anymore and they can’t get pregnant mothers anymore.”*



*“I don’t have to worry about them [the council] coming. (…) Not having the worry of coming home and they’re gone and just yeah, overall, happy.”*


The participants also suggested a feeling of fulfilment and satisfaction through their caring for the cats after receiving support from the program. This sense of fulfilment generally came from seeing the change in the cats’ circumstances and welfare, and being the person who facilitated that change. This is evidenced in the quotes below:


*“It’s really satisfying and brings a lot of joy to see that transformation in an animal.”*



*“These things [the cats] are so spoiled, so loved, and it’s like it’s nice to know that I played a part in that.”*



*”It makes me happy. Yeah, I know I’ve made a difference here.”*



*“…it’s really satisfying to see him get well to the point where he can play and he’s got that energy and a little bit of joy in life actually, that certainly wasn’t there [before}.”*



*“I’m really happy and satisfied that I was able to provide somewhere safe for them.”*


The caregivers further revealed they had a sense of pride given the outcome of the cats, and a feeling of empowerment now they had the knowledge and resources to act in the best interests of the cats. These feelings are explored in the quotes below:


*“Yeah, I love looking at them now. I thought, oh, all that hard work and now they’re looking really good.”*



**
*“*
**
*It’s helped to take that intent to do the right thing and turn it into some sort of an action because I feel like I can, yeah.”*



*“It’s made it easier for us to make a difference, not just in the cat’s lives, but in the community too.”*



*”Just simply having some knowledge was empowering.”*


Some of the caregivers also explained that they had experienced an improvement in their relationships with the cats after they had been sterilized, as the cats had become more friendly, calmer, and more willing to have contact with the caregivers. This improvement in the human–animal bond is shown in the following quotes:


*“The relationship has got better since they were desexed because before we couldn’t get near them.”*



*“So yeah, they just get a lot more friendly. (…) Yeah, it feels good that you can actually get beside them and give them a bit of a pat.”*



*“…the bond with them wouldn’t have grown into what it is without having them desexed.”*


#### 3.3.2. Social Capital

When given the opportunity to talk about the semi-owners’ and owners’ experiences after receiving support from the Community Cat Program, the theme of social capital arose. The interview data suggested an improvement in social capital, which is linked to quality of life, through two main sub-themes: social connections and community engagement. The cat caregivers indicated an improvement in their social connections after having the cats sterilized through the program, and discussed feeling that they had a support network that they could rely on if they needed help. This perception of social connectedness is shown in the following excerpts:


*“It gave us something to talk about quite a lot. Yeah. Yeah, because there was that common interest between us.”*



*“…probably a bit of marriage help too, because the Community Cat Program was there, I wasn’t turning to my husband and saying, OK, I need AU $100 per cat, you know, I need AU $1000 to go and desex these things and there would have been arguments and stuff like that.”*



*“I know that, through the local resident’s association, that there are people who are right behind the program. So, well, even if the program ended, there are some people who self-identified that I could reach out to if I needed.”*


The analysis of the interview data suggested that the cat caregivers were participating in community engagement, specifically by recommending the Community Cat Program, after receiving support from the program themselves. Some of the caregivers discussed that they were planning to actively engage with their local neighborhoods to trap and sterilize stray cats with the support of the program. The following quotes give context to this:


*“…I make sure that I say, oh, there’s this program that, you know, looks after community cats and stray cats, and you can get your cat desexed if you live in these areas and stuff like that. So, I make sure I get it out there.”*



*“[I was walking past] the front of our back neighbor’s property, there was a little cat and [I’m] like, oh, I haven’t seen that cat before you know, and I’m [wondering] what’s going on and then my partner happened to bump into her [the neighbor] down at the shops and she happened to mention there’s these strays. And then I gave her a call and I’m like, hey, do you wanna trap and things?**”***



*“I was thinking the other day of just doing a letterbox drop around to say like do you have a stray cat? (…) Are you aware of stray cats or anything like that? Get in contact with me, I’ve got the facilities where I can come and get them desexed and see if we can rehome them.”*


#### 3.3.3. Improved Cat Welfare

When asked about the welfare of the cats after receiving support from the program, the responses from the caregivers suggested an improvement in the cats’ welfare. Specifically, an improvement in their general health was discussed, as well as the fact that the cats now displayed play behaviors. These indicators of improved health and welfare are evidenced by the quotes below:


*“They [the cats] are more settled, they’re more relaxed, and I think it’s better for them, actually. (…) They have put on weight and they look really nice. You run your hand on them and they feel like velvet.”*



*“They [the cats] weren’t bad before, but in comparison, the condition is very, very good now and (…) their health is actually really good for what they eat, yeah.”*



*“Now they’re [the cats] just playing all of the time, running around all of the time and it’s… it’s really good to see that. It’s sort of like they’re enjoying their life a bit more.”*


#### 3.3.4. Perception of the Community Cat Program

When given the opportunity to discuss their experiences and perceptions of the Community Cat Program, the caregivers described having positive experiences with the program, which were articulated through four main sub-themes: feeling supported, educated, the program being flexible and easy to engage with, and having trust in the outcomes for the cats. Those who had previously reached out or had interactions with the council also compared their experience to that with the Community Cat Program, indicating having had a more positive experience with the Community Cat Program. Evidence of this is provided in the following excerpts:


*“I feel more confident with the situation and partly because the program gave me sort of information and insights about it, but also feeling that there’s some support, which is a level of support that I certainly didn’t feel from council.”*



*“[The Community Cat Program has] allowed us to make a difference and having that support under [us] has meant that I can now think of what else I can do in the community to help.”*



*“…just having somebody there to support you and understand why you’re doing it and also helping you try and achieve a better situation for them [the cats]. It’s really good.”*


Additionally, the cat caregivers suggested that they felt educated by the Community Cat Program, gained knowledge on how to handle cat-related situations, and were given an awareness of where to go if they had questions regarding the cats. The caregivers described this in their own words in the quotes below:


*“I suppose having a greater awareness of the presence of shelters and rescues around the place, who I might be able to call upon, even if it’s just for advice.”*



*“[The Community Cat Program’s Community Liaison Officer] is just there to help. (…) Because she explains things to you, and if I ask her a question she answers it. That’s all you need.”*


The flexibility and ease of the program was discussed by the cat caregivers, suggesting that the flexibility of the program gave those who could not miss work the ability to have their cats sterilized. As such, the caregivers perceived the program to be easy to engage in, with members of the Community Cat team appearing to build a good rapport with the caregivers. This sub-theme is given more context in the following quotes:


*“The ease of the program was fantastic and that was one of the big things that we thought was just great because we’re busy enough. Just all the little things like with [the Community Cat team] being able to come out here and pick up these cats and take them and deal with them and deliver them back (…) So that was—to us, that was probably a big thing too because our time is money as well.”*



*“I could go to work, knowing that the cats are being picked up and they’re being taken care of*
*, and then I’d come home and they’d be here and or I just need to set up a trap and just wait. So it was, it was really easy.”*


The cat caregivers indicated that they trusted the Community Cat Program and that the outcomes for the cats and the caregivers were going to be positive. Additionally, the knowledge that healthy cats were not going to be euthanized appeared to help build this trust, as evidenced in the quotes below:


*“I trust them [the Community Cat Program] heaps more, oh, yes. I have no problems. Because I know they’re going to do what’s best for the cat and what’s best for me, and—I trust them.”*



*”What I liked, was that yeah, she wasn’t going to be killed just because she was wasn’t able to be found a home.”*


### 3.4. Without Support from the Community Cat Program

The cat caregivers were asked during the interviews what they would have done if they had not received support from the Community Cat Program. The responses to these questions revealed that several of the caregivers would have surrendered the cats to the authorities, with some assuming negative outcomes for the cats. The caregivers used words like “devastated” and “sad” when describing how they felt about surrendering the cats. The perception of what may have happened without assistance from the Community Cat Program is shown through the following quotes:


*“Well, we would have to end up taking them to the council.”*



*“If the program hadn’t have been here, (…) I suspect I might have ended up dumping them with Council and they, I assume, therefore they [the cats] would need to pass health checks and behavioral checks and find an owner within a couple of days, before they’re killed so… Um which isn’t a great outcome for them”*



**
*“*
**
*If I haven’t got them desexed, and it kept continuing that my own cats were attacking me, I would have to give them up to protect myself and my cats. (…) I would have been devastated, the fact that I wasn’t able to provide basically a home for them, (…) it would have broken me to have to give them up.”*


### 3.5. Impact on Rural Participants

One interview was conducted with two participants who worked and lived on a farm, and some of the responses from their interview differed from those living in urban areas. We believe that these are important to report, and the farmers’ relationship with the cats and the impact of the Community Cat Program on the farmers are shown in the following section.

The results from the thematic analysis indicate that the farmers viewed the cats they cared for as working animals. They discussed having a problem with rodents, which had improved with the presence of the cats. This made the cats extremely important to the caregivers, and is described in their own words below:


*“We never ever have to rejoin a wire and basically that was most of what the technician was doing, was just coming back finding cut wires that had been chewed by the rats and mice and—so yeah, they’re a pretty important. We definitely don’t intend to get rid of them.”*



*“Like our dad came here nearly 70 years ago—there would have been rats and mice in the haysheds and in the sheds but now we don’t see any. We don’t even see a mouse.”*


The reduction in rodent problems reduced financial worries for the farmers by saving money from hiring electrical technicians and buying rodent poison. This is shown in the quotes below:


*“A couple hundred dollars a year worth of saving by not using [rodent] bait stations and stuff like that.”*



*“So, in that, it probably has a financial saving of somewhere between AU$3000 and AU$4000 a year [for technician time to fix wiring].”*


The farmers stated that without the Community Cat Program, they would not have been able to afford sterilization, and would have had to cull the cats to keep the numbers under control, which they indicated they did not feel positively about. This is shown in the excerpts below:


*“The greatest benefit to us was because we’re struggling dairy farmers, we didn’t have to pay to get it done and that was—yeah, that was great—that was the big thing—and, to tell you the truth, in the situation that we were in, especially at that time, we wouldn’t have even given it a thought because we couldn’t afford it.”*



*“It could well have got to that point where we might have had to cull them or something (…) it wouldn’t have been a real good option.”*



*“We would have had to cull them somehow, I’d say. (…) We had to put down animals in the droughts, so we don’t really want to do it, if we don’t have to.”*


## 4. Discussion

We conducted a qualitative analysis of the interviews with people who were caring for multiple cats, with the aim of exploring their relationship with the cats and the impact of an assistive-centered management strategy on the caregivers’ psychological well-being and quality of life. The key findings of our research are the strength of the bond between the caregivers and the cats they cared for, and the positive impacts of an assistive-centered management strategy on the cats’ nuisance behaviors, the caregivers’ quality of life and social capital, and the cats’ welfare. Additionally, we found that the Community Cat Program was perceived more positively when compared to agencies such as the council.

### 4.1. Relationship with Cats

The cat caregivers described the strong emotional bond they had with the cats they cared for, regardless of perceived ownership, with several of the participants describing the cats as family members or children—“they’re my kids”. Additionally, the caregivers discussed the calming effects of spending time and interacting with the cats, and the benefits this had on their mental well-being, saying things like, “if I’m really down, I’ll actually come out here and cuddle them”. There is an extensive literature that focuses on the bond between pets and their owners and the benefits of this bond [44,45,46,47,48]. Additionally, studies have shown that viewing pets as family suggests a strong emotional attachment to the animal [49,50]. Our study demonstrates that those feeding and caring for stray cats also have a strong bond, and feel similar benefits to their mental well-being as those who own companion animals, which is consistent with other studies [19,21,22,26]. A recent study, which used the Comfort from Companion Animals Scale, found the strength of the bond between cat caregivers and their cats (mean 39.6, SD 5.9) was almost identical to the bond cat owners felt with their pet cats (mean 39.6, SD 4.8) [22]. The strength of the bond was also evidenced by the amount of money the caregivers spent on the cats, despite living in low socioeconomic areas, with many spending more than the average spent on owned cats [51]. The level of perceived ownership appears to have little impact on the strength of the human–animal bond and the associated benefits. Furthermore, the value individuals place on free-roaming cats may improve the case for the cost-effectiveness of a free sterilization program [52]. These findings are important when considering animal management strategies, because the killing of the cats being cared for by multi-cat carers has been shown to have long-term adverse psychological impacts due to the sudden severing of this strong bond [19]. Therefore, when estimating the economic case for urban cat management, the value that individuals place on free-roaming cats is important to consider [52], as well as the cost of severing that bond.

### 4.2. Nuisance Behaviors

Before becoming involved with the Community Cat Program, the caregivers spoke about the issues they had with the cats surrounding nuisance behaviors, such as “they spray and they poop, and they smell” and “they have fights at night, and that sets off our dog”. These nuisance behaviors appeared to impact their quality of life, due to the negative effects on the caregivers’ physical environment and the potential health risks, suggested by statements such as “there’s a health implication with so much feces around” and “it keeps me awake during the night.” When asked about their experiences after the program, the participants made no mention of the issues surrounding nuisance behaviors, suggesting that the situation had improved for the caregivers. This perceived decrease in nuisance behaviors is consistent with the findings in the literature, whereby an increase in sterilized cats through a low-cost sterilization program led to a decrease in cat complaints, impoundments, and euthanasia [53]. Additionally, the World Health Organization (WHO) has indicated in their quality of life assessment scale, that an individual’s or population’s environment is the most important indicator of quality of life [54]. Moreover, less sleep has been associated with lower levels of happiness and quality of life [55,56]. The perceived reduction in nuisance behaviors is important when considering animal management strategies, as a reduction in nuisance behaviors is likely to reduce complaints made to the council about urban free-roaming cats, whilst having a positive impact on the cat caregivers’ well-being.

### 4.3. Quality of Life

Before receiving support from the Community Cat Program, issues surrounding the cats were perceived to be impacting the cat caregivers’ quality of life. Many of the caregivers experienced a strain on their marriages, friendships, or relationships with their neighbors. Several of the caregivers were worried about the cost of caring for the cats and indicated that cost was a barrier to the caregivers sterilizing their cats, stating “I couldn’t afford the desexing” and “I’m not prepared to pay that amount of money”. Additionally, being faced with a problem, but not having the knowledge or the means to do anything about it gave caregivers a sense of powerlessness. The feeling of wanting to do the right thing but not being able to do it due to external barriers has been shown to cause moral distress, leading to negative psychological impacts [57]. This is consistent with our findings, as the participants indicated feelings of powerlessness, e.g., “there’s a problem and not feeling that I had the power to do something about it”. Additionally, financial worries can have a negative psychological impact on an individual and lead to a decrease in quality of life [58,59]. These negative impacts can be diminished by social support [59,60]. However, the responses during the interviews suggested that due to the strain on their social relationships, the caregivers had no way of ameliorating their negative emotions, leading to a further negative impact on their quality of life.

This differs from the responses caregivers gave when asked about their experience after engaging with the Community Cat Program. The participants discussed feeling less worried and were relieved that they had received support, stating “I feel really good” and “we don’t have to worry”. Additionally, the caregivers explained that they felt a sense of empowerment, as they had been given the resources and knowledge to take action, with one stating that “simply having some knowledge was empowering”. When discussing the change they had seen in the overall situation with the cats, the caregivers spoke about finding satisfaction in being able to make a difference, with one stating “I’m really happy and satisfied”. This indicates a sense of fulfilment, and the participants discussed the sense of pride they felt that the cats had a positive outcome, such as “We enjoy showing them to people”. These positive emotions and sense of accomplishment and life satisfaction indicate an improvement in their quality of life [54,61,62]. Additionally, positive emotions, such as empowerment, can influence behavioral changes [63,64]. This is important because positive behavioral changes are key to the success of urban cat management programs [36,65].

Our findings have highlighted the fact that after receiving assistance from the Community Cat Program, the cat caregivers felt an improvement in their quality of life. This is significant, as it demonstrates that assistive-centered management is beneficial to the caregivers’ psychological well-being and is aligned with a One Welfare approach which optimizes the well-being of humans, animals, and the environment.

### 4.4. Social Capital

When the caregivers were asked about their experiences after receiving assistance from the Community Cat Program, they discussed an improvement in their social capital. The cats were facilitators of social contact, improving and facilitating caregivers social relationships, e.g., “It gave us something to talk about”. Additionally, engaging with the program seemed to have helped enable community participation, by giving the caregivers resources, such as the knowledge and equipment to engage with their local community. The caregivers discussed giving their neighbors information about the program, and some talked about plans to work within their local community to help trap and sterilize free-roaming cats. Whilst there appears to be no research on the impacts of semi-ownership on social capital, our findings are consistent with the literature on pet ownership and social capital, which indicates that pet owners have higher social capital than non-pet owners [45,66]. Moreover, increased social capital is strongly associated with an increased quality of life [67,68]. Cat caregivers are generally women [9,21,26,69] who live in low socioeconomic areas [26,69]. This demographic has been shown to have lower social capital, leading to inequalities in health [70,71]. Our findings suggest that assistive-centered cat management can improve social capital, providing members of the community who are likely to experience lower levels of social capital with the resources with which to build social and community connections. Additionally, the increase in community engagement is not only beneficial to the caregiver’s social capital and quality of life, but is vital for the success of urban assistive-centered cat management programs [64,65].

### 4.5. Cat Welfare

Before receiving support from the Community Cat Program, the participants expressed concern over the welfare of the cats they cared for. They mentioned that the cats were in poor health and looked “really skinny”, and suggested that the cats were frequently getting injured due to fighting, e.g., “they had a lot of scarring because of the fighting”. In contrast, after the cats had been sterilized through the Community Cat Program, the caregivers indicated an improvement in their welfare, e.g., “they are in such good condition” and “they’re playing all of the time”. An improvement in cat welfare after sterilization is well documented [72,73] and consistent with our findings; several studies have shown a decrease in aggressive behaviors after cats have been sterilized [74,75], leading to a decrease in injuries. Overall, this shows that sterilization through a One Welfare approach improved the health and welfare of the cats being cared for.

### 4.6. Perception of Support

The participants discussed their perception of support from agencies before engaging with the Community Cat Program. The caregivers who had contacted various agencies, such as the local council, and asked for help illustrated that they felt unsupported, e.g., “there was just nobody there to help”. They discussed feeling that without support they did not feel like they could do anything about the issues surrounding the cats they were caring for, e.g., “it still felt out of my grasp”. The caregivers who had worked closely with the council regarding complaints discussed feeling angry about how the council dealt with the complaint, e.g., “I felt really angry”, and relayed their feelings of fear about the possibility of future interactions with the council, using words such as “dread”. Several of the participants indicated that they did not trust the various agencies and feared the cats would be taken away and euthanized; therefore, they did not engage with them, stating “I didn’t want them to take my cats away” and “I don’t want to engage with the council”. Negative interactions with and distrust of authorities have been shown to be barriers to engagement with community services, particularly in low socioeconomic areas [76,77], which is consistent with our findings. Additionally, these interactions and lack of support create negative emotions for caregivers, which have been shown to lower their quality of life [59]. These findings are similar to those of previous research conducted on enforcement-centered cat management [19]. The negative perception of the council may be perpetuated by animal control officers not being able to provide the appropriate support to communities due to policy and financial restrictions. One study indicated that animal control officers faced barriers to engaging with the community, due to a lack of funding, staff, and resources, even though many animal control officers perceived programs like the sterilization of stray cats (TNR) as good for the community [78].

The caregivers’ perceptions of the support they received from the Community Cat Program differed from their perceptions of agencies, such as the council. The caregivers relayed feeling supported and educated, e.g., “there was somewhere I could go to get the help”, and stated the trust they felt in the program, “I trust them”. Ongoing communication from the Community Cat Team to determine if additional assistance, such as cat food or cat litter, was required is likely to have helped build these feelings of support and trust. Additionally, the caregivers appreciated the flexibility and ease of the program, stating: “the ease of the program was fantastic”. This appeared especially helpful for those who could not take time off work to get the cats sterilized, e.g., “I could go to work” and “our time is money”. For low-income workers, taking time off work can result in the loss of income, and is an additional obstacle to getting cats sterilized. The program generally led to positive emotions, with participants using words such as “gratitude” and “confidence”. Similar to other studies, our findings indicate that positive engagement with communities, including the provision of educational materials, can build trust [64,65]. This has been suggested to increase the reporting of unowned cats for sterilization and to reduce the number of unwanted litters [65]. Moreover, community engagement is likely to improve social capital [45] and, coupled with the positive emotions experienced by the caregivers due to the assistance from the Community Cat Program team, suggests a positive impact on their quality of life [54,61,62,67,68].The difference in the perceived level of support from agencies, such as the council, and the perception of the support received from the Community Cat Program is evident from our findings. This is crucial when considering animal management strategies, because an assistive-centered approach appears to build trust with communities, creating positive behavioral changes, which are essential to the success of urban cat management programs [36,65]. Additionally, an assistive-centered management approach encompasses the One Welfare philosophy by positively impacting the caregivers’ quality of life. The barriers animal control officers face to engaging with an assistive-centered approach should be investigated, as these may hinder a One Welfare approach to urban cat management.

### 4.7. Without the Community Cat Program

When the participants were asked what they would have done had the Community Cat Program not provided assistance, most of the caregivers stated that they would have surrendered the cats to the council or an animal rescue shelter, e.g., “we would have to end up taking them to the council.” The caregivers also indicated that they assumed the cats would have been euthanized, and used words such as “devastated” and “sad” to express how that would have made them feel. Most cats surrendered by the public to shelters are strays [79,80], and one study found that over a third of stray cats were associated with a caregiver (semi-owner) for more than a month before being surrendered [79]. Our findings suggest that an assistive-centered cat management strategy focusing on semi-owned cats may decrease the number of stray cats and kittens entering shelters and municipal facilities. Additionally, the responses from the interviews indicate that without the assistance of the Community Cat Program, the caregivers may have felt a negative impact on their quality of life. Overall, this indicates that an assistive-centered management strategy not only improves the outcomes for caregivers and cats, but could also reduce the workload for animal management officers and shelter staffs.

### 4.8. Impact on Rural Participants

Our findings show that the relationships with the cats and the impacts of the Community Cat Program differed for the rural participants. The farmers who were interviewed indicated that they saw the cats as valuable working animals, using words such as “important” to describe them. Additionally, the presence of the cats improved their rodent problem and, therefore, lowered operation costs: “we don’t even see a mouse” and “a financial saving of somewhere between AU $3000 and AU $4000 a year [for technician time to fix wiring]”. This is important, as the farmers noted that they struggled financially, stating “we’re struggling dairy farmers”. This decrease in operational costs could also decrease their financial worry and positively impact the farmers’ quality of life [58]. Furthermore, the farmers discussed not being able to afford sterilization, indicating that without assistance from the Community Cat Program, they would have had to cull the cats to keep their numbers under control. When discussing this management option, the farmers appeared to feel negatively about it: “we don’t really want to do it if we don’t have to” and “it wouldn’t have been a real good option”. No research has been conducted on the impacts of farmers culling working animals, such as cats or dogs. However, the culling of livestock during epidemics has been shown to lead to negative emotions, substantial psychological distress, and post-traumatic stress disorder in farmers [81,82]. Drawing from the farmers’ responses and the literature, it could be assumed that the culling of the cats would have negatively impacted the farmers’ quality of life. There needs to be further research conducted on the relationship farmers have with the cats they care for, as well as the impact of assistive-centered cat management on farmers’ psychological well-being. However, our findings provisionally suggest that using a One Welfare approach to cat management had a positive impact on the farmers’ quality of life. Additionally, decreasing the number of kittens born would be expected to decrease the environmental impacts, such as native wildlife predation and contamination by toxoplasmosis oocysts, because cats under one year of age are the predominant source of oocysts [83,84].

### 4.9. Limitations

We recognize the small sample size of this study; however, the aim of this study was to gain an in-depth understanding of the bonds caregivers have with the cats they care for, and the impact of an assistive-centered cat management strategy. To gain this in-depth understanding, a smaller sample size was necessary to gather rich and detailed data within the time and financial constraints. Additionally, the semi-owners of multiple cats are only a small proportion of cat semi-owners. The Community Cat Program has sterilized over 2600 cats in 3 years, and this included cats from 13 multi-cat sites, which had a median of seven cats per site. Therefore, our sample size of 11 multi-cat semi-owners is an appropriate representation of this small population. Previous qualitative research of a similar nature has used a small sample size to gain an in-depth and rich understanding of a topic [19]. We hope that our findings can act as a foundation for future research to build upon, as well as a resource and catalyst for other communities seeking to implement an alternative strategy for urban cat management based on a One Welfare approach.

## 5. Conclusions

This study has demonstrated the positive impacts of an assistive-centered cat management strategy on caregivers’ psychological well-being and quality of life, and a positive impact on cats’ health and welfare. Additionally, we have shown the caregivers’ negative perception of agencies like the municipal council, and the potential this has to hinder urban cat management efforts. Moreover, the results have provided further evidence of the strength of the bond between caregivers and the cats they care for, regardless of perceived ownership. Implementing an assistive-centered management approach not only improves human well-being and cat welfare, but also creates a positive relationship between the community and the agencies involved. This can lead to improved reporting of cats requiring sterilization, which is key to the success of an assistive-centered management program. Therefore, this One Welfare approach to urban cat management has the potential to reduce the free-roaming cat population, their impact on wildlife, and nuisance complaints, and to reduce council and shelter impoundments and costs. We hope our findings will provide evidence to local governments and welfare agencies of the beneficial impacts of an alternative, assistive-centered urban cat management strategy. We hope that our study can be used as a pilot study, and that the themes which arose during the interviews can be used as a base for future studies to build upon, and as a global resource for the development of effective urban cat management programs based on a One Welfare philosophy. 

As one of the participants commented: “…cases where there’s 10 or 15 cats on one property, if the council can deal with it in the way that we dealt with it, they’re not going to end up with 10 or 15 cats at the pound. They’re actually going to end up with these cats back where they are, not causing any trouble to anyone. (…) It’d be a fantastic thing for the councils to be able to take it on board. It’s just a good outcome for everybody.”

## Figures and Tables

**Table 1 animals-13-03423-t001:** Major themes and sub-themes with context examples from interview transcripts.

Theme	Sub-Themes	Context Examples
Relationships with Cats	Strong bond Calming/joy Investment Responsibility	“We’d never be without them.” “They’re very calming, you know, it’s really nice and they love being stroked.” “So, we had to buy (…) a big pen, and they sleep in there.” “If I don’t feed them, nobody else is gonna feed them.”
Without the Community Cat Program	Surrender cats Sad/devastated	“Yeah, I would have had to surrender them.” “Yeah, I was gonna take them all down to the pound.” “They probably would have been put down, you know, and that would have made me really sad, really.”
Before the Community Cat Program		
Human Quality of Life	Strains on social relationships Powerlessness and lack of knowledge Worried	“Neighbours used to throw letters and that over the fence, [saying] do something with your cats (they weren’t my cats).” “I was worried about having repeated unwanted litters. Not knowing how to deal with the fact that there are all these stray cats hanging around.” “I was worried about how much it would be to desex like 20 cats.” “I couldn’t afford desexing or the microchipping in the position that I was in.”
Nuisance Behaviors	Noise Defecation Unwanted litters	“[There] used to be a lot of strays [that] used to come round here and used to annoy me (…) it kept me awake during the night.” “Biscuit would (…) terrorize everyone, he would force himself I suppose onto the females, and it was just a lot of fighting, a lot of fighting.” “They were just defecating everywhere and it was not pleasant. Yeah, it didn’t smell nice either.” “I thought, no, it’s just not good. It’s not fair on them for having so many litters. It’s not fair on them and it’s not fair on us.”
Cat Welfare	Poor condition Injuries Lack of play Concern for safety	“She looked really skinny. So, we were concerned for her.” “…he was pathetically skinny.” “They used to fight. (…) One or two of them would come back with just a heap of hair missing or whatever.” “…their hair had looked like they weren’t cleaning themselves properly. I think they probably were too stressed.”
Perception of Authorities	Lack of support Negative interactions Fear of outcome for cats	“We kept asking for help, and there was just nobody there to help.” “I never took anything back once I found out what they [the Council] do. Healthy cats, they’re just put to sleep, and I thought, no.” “I suppose I was a bit wary of Council.” “[I] strongly suspected that if Council took the animals, their outcomes probably wouldn’t be great.”
After the Community Cat Program		
Human Quality of Life	Less worried Fulfilment and satisfaction Pride and empowerment Improved relationship with cats	“Just for me, just getting it done knowing that my cats are safe is… is great.” “Yeah, we’re all getting sleep like the cats aren’t overwhelming us.” “It feels really good to see her do that. (…) Now she wants to play with us and chase us around the yard.” “We enjoy showing them to people. We can show them to people, and we were never ashamed of them, but now, we definitely aren’t ashamed.”
Social Capital	Improved social connections Community engagement	“I know there is a community that I can go to if something’s wrong or if I need help with them [the cats].” “…now that they’re desexed and microchipped, [my friend is] so much more, you know, chill with [accepting] them.” “All the ones that I’ve rehomed, I’ve always connected them in with the cat team and RSPCA so they’ve [the new owners] all done the responsible thing too and gotten them desexed.” “I started thinking, well, maybe there’s other cats out there that I could trap, work with, get desexed.”
Improved Cat Welfare	Improved health Play behaviors	“If you see those cats now, the mother cats, they are in such good condition. I cannot get over it.” “But since it’s all been sorted, Poppy now has the zoomies of an afternoon (…) like she’s playing, and we’ve never seen her play.”
Perception of the Community Cat Program	Supported Educated Flexibility and ease Trust	“I know they’re the ones to go to now [Community Cat Program team], because they are the ones that will support me through it. So, they’re pretty much the only ones I refer to now.” “There was somewhere I could go to get the help, that it wasn’t going to bankrupt me.” “I feel a lot of gratitude to the program for… for the support and for helping… I suppose for educating.”

## Data Availability

Most relevant data are reproduced in the text.

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
