# Peer review of "Solutions-Based Approach to Urban Cat Management—Case Studies of a One Welfare Approach to Urban Cat Management"

_animals, 2023, doi:10.3390/ani13213423_

Round 1

Reviewer 1 Report

Comments and Suggestions for Authors

Line 98/99. Need to add some sort of a transition as you are talking about a community in Brisbane and then suddenly comparing it to Ontario. So a transition of some sort explaining how you can compare the vastly different communities might help smooth this out 

Line 100/101 need to Identify who or what is one welfare.  No discussion explaining this organization or program

Line 112/113 explain assistive management strategy 

Line 117/188 explain shelter based return to field

Line133 explain the testing  also what are the numbers of cats euthanized yearly quantify the problem

Line138 rework the sentence - too long - reader gets lost

Methodology- explain how data was kept confidential and privacy of human subjects was protected.  What was the process of theme development in the reading of transcripts  How do you think this small number of interviews related to the population of feral cat caregivers 

What does the community cat program offer besides spay/neuter?  Do the regularly contact caregivers?  If so does the multiple times contacted build a stronger relationship with caregivers which might help to explain more positive feelings in comparison to the governmental office?  How much does social class play into negative feelings towards government officials?

Reviewer 2 Report

Comments and Suggestions for Authors

This is a well-designed, carefully researched study of a topic with considerable significance. I appreciate the detailed explication of the research methods. The literature review is comprehensive yet focused. The paper is well organized and thoroughly cited. I have no substantive edits to recommend, but I will make two suggestions. 

First, you might consider adding a few sentences--no more than that--emphasizing the potential application of the findings to other communities seeking cat management strategies. Of course, you say as much already, but I would like to see the claims be more emphatic, perhaps presenting this as a pilot study for future global efforts. 

Second, you might also consider adding a mention about the problematic nature of the term "feral." For instance, Kristine Hill's work (one paper cited below) discusses how people use the term to wield power. The paper also happens to use the case of Australian cats. 

Hill, Kristine, Michelle Szydlowski, Sarah Oxley Heaney, and Debbie Busby. 2022. "Uncivilized behaviors: How humans wield “feral” to assert power (and control) over other species." Society & Animals (1)1-19.  https://doi.org/10.1163/15685306-bja10088

Reviewer 3 Report

Comments and Suggestions for Authors

This manuscript describes the impressions of a very limited number of surveys (possibly 5 interviews with a total of 11 participants) on the effects of a desexing and vaccination program for stray cats on the mental health of 'semi-owners'.

This study may be of interest to a psychology journal but because no evidence of effects on animal welfare or animal management are provided, its relevance to Animals has not been justified. 

Of note, Human Ethics permission was granted (as required for public questionaires) but no Animal Ethics permission was cited for the interventions to cats that formed the basis of this study (Animal Ethics permissions are typically a pre-requisite for research - and especially publication - involving the experimentation with animals)

Similarly, although not described specifically, it appears that these unowned cats (justification will need to be provided for use of the ambiguous and dubious new term 'semi-owned') were released back onto the streets/farms, but no indication of whether this was approved or allowed by State or local laws. 

Whilst feeders of stray cats obviously form bonds with these cats and resent and are stressed by authorities to aim to minimise the nuisence and animal welfare (both cats and wildlife) of free-ranging cats, this study provides no data on the impact of the legality or ethics of the interventions described, nor the effect of such intervention on cat welfare, population size and impacts of these cats.

Improved management of unowned cats continues to present challenges for many stakeholders, including the welfare of cats. The perceptions of a few stray cat feeders (cat 'caregivers') is of some importance but, in a scientific journal devoted to improving animal management, is not as relevant as measured responses of cat populations, cat health, nuisence reports and wildlife populations to experimentation that is preferred by feeders of these unowned cats. Similarly, the economic arguments of cost savings of  cat-mediated rodent control should be compared with costs of appropriately owning a cat and alternative rodent control techniques

Round 2

Reviewer 1 Report

Comments and Suggestions for Authors

Thank you for addressing suggested changes